# Clinicians Are Not Able to Infer Parental Intentions to Vaccinate Infants with a Seasonal Influenza Vaccine, and Perhaps They Should Not Try: Findings from the Pediatric Influenza Vaccination Optimization Trial (PIVOT)–IV

**DOI:** 10.3390/vaccines10111955

**Published:** 2022-11-18

**Authors:** William A. Fisher, Vladimir Gilca, Michelle Murti, Alison Orth, Hartley Garfield, Paul Roumeliotis, Emmanouil Rampakakis, Vivien Brown, John Yaremko, Paul Van Buynder, Constantina Boikos, James A. Mansi

**Affiliations:** 1Department of Psychology, Department of Obstetrics and Gynaecology, Western University, London, ON N6A 3K7, Canada; 2Département de Médecine Sociale et Préventive, Faculté de Médecine, Institut Nationale de Sante Publique du Québec and Université Laval, Québec City, QC G1V 5B3, Canada; 3Dalla Lana School of Public Health, University of Toronto, Toronto, ON M5T 3M7, Canada; 4Fraser Health Authority, Vancouver, BC V3T 0H1, Canada; 5The Hospital for Sick Children, University of Toronto, Toronto, ON M5G 1X8, Canada; 6Eastern Ontario Health Unit, Cornwall, ON K6J 5T1, Canada; 7JSS Medical Research, Montreal, QC H4S 1N8, Canada; 8Department of Family and Community Medicine, University of Toronto, Toronto, ON M5G 1V7, Canada; 9The Montreal Children’s Hospital, Montreal, QC H4A 3J1, Canada; 10Department of Pediatrics, McGill University, Montreal, QC H3A 0G4, Canada; 11School of Medicine, Griffith University, University of Western Australia, Perth, WA 6009, Australia; 12Seqirus, Montreal, QC H9H 4M7, Canada

**Keywords:** influenza, childhood vaccination, parental acceptance, vaccine hesitancy, education

## Abstract

This prospective cohort survey evaluated the concordance of clinicians’ perceptions of parental intentions and parents’ actual intentions to vaccinate their infants against influenza. During a routine healthy baby visit, clinicians provided parents with information about influenza, children’s vulnerability to influenza, and nonadjuvanted and adjuvanted trivalent influenza vaccines (TIV and aTIV, respectively). Before and after the clinician–parent interaction, parents were surveyed about their attitudes, their perceptions of support from significant others, and the intention to vaccinate their infant with aTIV. Clinicians were asked about their perception of parents’ intentions to choose aTIV for their children. These assessments included 24 clinicians at 15 community practices and nine public health clinics, and 207 parents. The correlation coefficients of the clinicians’ assessment of parents’ intention to vaccinate were 0.483 (*p* < 0.001) if the vaccine was presented as free of cost, 0.266 (*p* < 0.001) if the cost was $25, and 0.146 (*p* = 0.036) if the cost was $50, accounting for 23%, 7%, and 2% of the variance in parental intentions, respectively. The clinicians were poor at predicting parental intentions to immunize, particularly when cost was involved. Information on vaccine options and influenza infection should be provided for every eligible patient to allow parents to determine if the vaccine is appropriate for their child.

## 1. Introduction

Children represent approximately 13% of the population infected with influenza globally and are at elevated risk of influenza-related complications [1,2]. Worldwide, influenza-related mortality ranges from 2.1 to 23.8 per 100,000 population among children younger than 5 years, and the highest percentages of hospitalizations and death are among children younger than 2 years [3].

Seasonal vaccination against influenza is recommended for certain subpopulations, including children aged 6 through 23 months, that account for a large proportion of influenza-related hospitalizations and deaths [4,5]. In Canada, three types of influenza vaccines are currently licensed for use in children of 6 through to 23 months of age: trivalent inactivated influenza vaccines (TIV), quadrivalent inactivated influenza vaccines (QIV), and MF59-adjuvanted trivalent inactivated influenza vaccines (aTIV; Fluad^®^, Seqirus UK Limited). aTIV has been shown to elicit an earlier, stronger, broader, and more persistent immune response in children when compared to nonadjuvanted TIV [6,7], particularly when evaluated across pediatric-relevant correlates of protection [8].

Despite the significant morbidity and mortality caused by influenza each year, as well as the availability of safe vaccines, low influenza vaccine uptakes within high-risk groups remain a global challenge and contribute to the burden of disease [9]. While parental vaccine concerns and vaccine misinformation circulating in social media and other outlets play an important role in parents’ immunization intentions [10], studies have shown that physician recommendations are the single most important facilitator of vaccine uptake in children of 6 through to 23 months of age [11,12,13].

At least anecdotally, clinicians often report that they do not offer vaccines with the same intensity of recommendation or even at all if they perceive that patients (or their parents) may be uninterested or unwilling to accept the vaccination in question. The ability of clinicians to correctly infer their patients’ readiness to accept seasonal influenza vaccination, however, remains largely unstudied. The objective of this research was to evaluate the concordance, or lack thereof, of clinicians’ perceptions of parents’ intentions to vaccinate their infants and the actual parental intentions to do so with a novel seasonal influenza vaccine. An additional aim of this study was to determine whether the accuracy of clinician perceptions of parental intentions to vaccinate may deteriorate when cost is involved, as in the case of a novel, approved, but not yet publicly funded vaccines. Accordingly, we assessed the accuracy of clinician perceptions of parental intentions to vaccinate infants with publicly funded influenza vaccines, as well as with vaccines that cost increasing amounts at parental expense.

## 2. Methods

### 2.1. Study Design and Population

This was a prospective cohort survey design conducted during the 2015–2016 influenza season. A complete description of the study design can be found in the PIVOT-I study report by Fisher et al. Briefly, the study population consisted of the parents of infants aged 6 to 23 months of age who were presented for a scheduled healthy baby visit. During the consultation with clinicians (physicians in community settings or nurses in a public health setting), information about influenza and aTIV was provided to parents (Figure 1). Parents were administered surveys by research nurses before and after the clinician interaction.

### 2.2. Study Procedures

After the clinician interaction, parents were asked about their intentions to vaccinate their infant with aTIV if it were provided free of charge, if it cost $25 and if it cost $50 and were asked to report their perceptions of the strength of the clinicians’ recommendation of vaccination with aTIV. Clinicians were asked, separately, about their perception of the parent’s intention to have their infants vaccinated with aTIV if free of charge, $25, and $50.

Parents’ intentions to vaccinate their infants with aTIV were assessed with a 7-point Likert scale where 1 = “Strongly agree” and 7 = “Strongly disagree” that the parent intended to vaccinate their infant with aTIV. The clinicians’ assessments of the parental intention to vaccinate with aTIV were assessed similarly with 7-point Likert scales where 1 = “Strongly agree that the parent intends to vaccinate their baby with the adjuvanted seasonal flu vaccine” and 7 = “Strongly disagree that the parent intends to vaccinate their baby with the adjuvanted seasonal flu vaccine.” Parental perceptions of the strength of the clinician’s recommendation to vaccinate with aTIV were assessed with a 7-point Likert scale where 1 = “Strongly recommended” and 7 = “Strongly discouraged.” This study was conducted prior to the availability of aTIV and focused on the parental intentions and clinicians’ assessment of parental intentions to opt for the vaccine when it became available. The research protocol received ethics approval from the Western University Health Sciences Research Ethics Board, the Fraser Health Research Ethics Board, and IRB Services.

### 2.3. Statistical Analysis

Correlation coefficients and corresponding 2-tailed measures of significance were calculated to assess the association of the clinicians’ perceptions of parental intentions to vaccinate their infants with aTIV and parental intentions to vaccinate per se. Correlation coefficients also assessed the relationship between the clinicians’ assessments of parental intentions to vaccinate their infants with aTIV and parents’ perceptions of the strength of the clinicians’ recommendations to do so.

## 3. Results

### 3.1. Sample Characteristics

A total of 18 community practice and public health clinics across Canada participated; 15 community practice clinics enrolled 136 parents, while three public health clinics enrolled 71 parents (N = 207). The baseline demographics and characteristics are presented in Table 1.

### 3.2. Clinician Perceptions of Parental Intentions

Healthcare providers were generally poor at predicting parental intentions to vaccinate their children with aTIV (Table 2). Clinician perceptions were most concordant with parental intentions in relation to parental intentions to accept the vaccine “when it becomes available” (r = 0.60, *p* < 0.001) and when the vaccine was presented as free (r = 0.483, *p* < 0.001). Even in the context of these statistically significant correlations, we noted that the clinician’s perceptions only accounted for 23% to 36% of the variance in parental intentions. When adding the cost of the vaccine, as is the case with approved but unfunded vaccines, the clinician’s concordance with parental intentions deteriorated. The clinician’s perceptions of the parental intentions to vaccinate accounted for only 7% of the variance if the vaccine cost $25 (r = 0.266, *p* < 0.001) and for only 2% of the variance if the vaccine cost $50 (*r* = 0.146, *p* < 0.02).

In a related analysis, it was determined that the clinicians’ perceptions of parents’ intentions to vaccinate their infants were associated with the parents’ perceptions of the strength of the clinicians’ recommendations to do so (r = 0.324, *p* < 0.001).

## 4. Discussion

These results shed light on the accuracy of clinicians’ perceptions of parents’ interests in vaccinating their children against influenza. Clinicians’ perceptions of parental interest in vaccinating infants with aTIV accounted for substantially less than half of the variance in the parents’ actual intentions to do so. When cost was involved, as is the case with approved but not publicly funded vaccines, the clinicians’ assessments of parental intentions to vaccinate their infants accounted for a very small amount (2% to 7%) of the variance of the parent’s actual intentions to vaccinate. Moreover, the clinicians’ assessments of the parent’s intentions to vaccinate their infants were associated with the parent’s perceptions of the strength of clinicians’ recommendations to vaccinate their infants. While this finding is correlational, and causality cannot be determined, it is possible that the clinicians who perceived parents’ interest in vaccinating their infants to be weaker gave correspondingly weaker recommendations to vaccinate. It is also possible that the clinicians who gave weaker recommendations to vaccinate influenced the parents to form weaker intentions to vaccinate their infant or that parents with weaker intentions to vaccinate were inclined to perceive clinician recommendations as relatively weak. All of these possibilities are worthy of further investigation.

Clinician recommendations are widely regarded as an important influence on vaccine acceptance. Our findings indicate that clinicians cannot accurately predict the parental acceptance of infant vaccination against seasonal influenza with a particular vaccine, especially if cost is involved. Our findings also suggest that parents who perceive weaker clinician recommendations of vaccination express weaker intentions to vaccinate their infants. While further research is needed—among the limitations of this research, we focused on intentions to use a vaccine that was not yet available, not on vaccination per se—and it seems clear that clinicians are not on particularly solid ground when attempting to gauge parental interest in vaccinating their infants. Accordingly, it would seem most reasonable to provide vaccine-related and preventable disease–related information to all parents of eligible infants so as to permit them to make vaccine decisions that are independent of clinician assumptions about parental acceptance. This seems to be especially important in the case of approved but unfunded vaccines because the clinician assessment of parental interest was weakest in this setting. It would also seem important to untangle the association we observed between the clinician perception of weaker parental intentions to vaccinate and the parental perception of weaker clinician recommendations of vaccination, given the importance of clinician recommendation influence on vaccine uptake.

## 5. Conclusions

Clinicians were poor at predicting the parental intention to immunize with aTIV. Information on the vaccine and the disease should be provided to every eligible patient to allow parents the option to determine if the vaccine is appropriate for their child.

## Figures and Tables

**Figure 1 vaccines-10-01955-f001:**
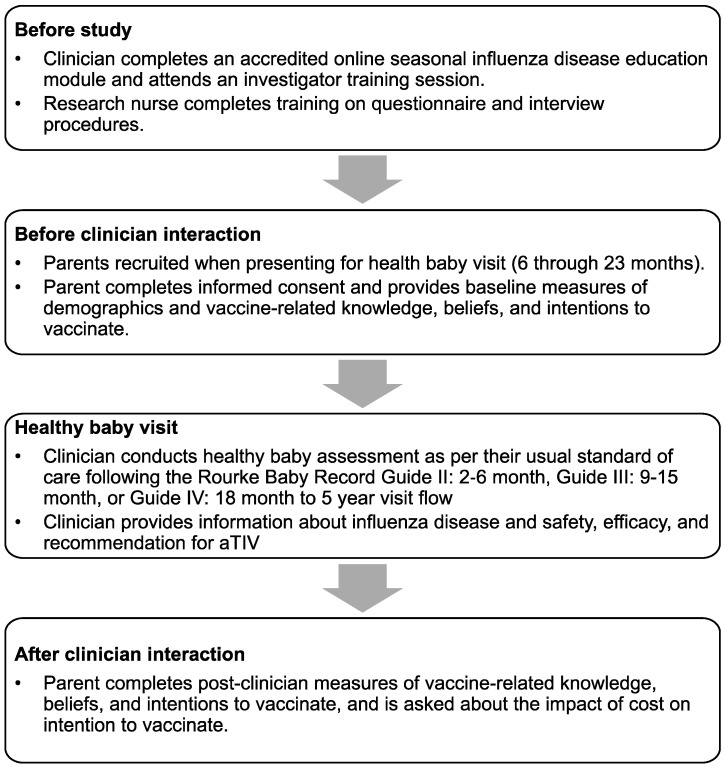
Study recruitment procedures and protocol flow.

**Table 1 vaccines-10-01955-t001:** Sociodemographic characteristics.

Characteristic	Survey Population (N = 207)
Mean age, (range)	
Parents (years)	33 (17–54)
Children (months)	13.5 (6–24)
Female sex, *n* (%)	
Parents	172 (83.1)
Children	101 (48.8)
Highest educational level attained by parent, *n* (%)	
University (bachelor’s degree or higher)	106 (51.2)
Community college, technical college, or trade school	64 (30.9)
High school or equivalent	35 (16.9)
Primary school	2 (1.0)
Parental race and ethnicity	
White	133 (64.3)
Asian	46 (22.2)
Native American	5 (2.4)
Black	6 (2.9)
Other	17 (8.2)

**Table 2 vaccines-10-01955-t002:** Correlation between clinicians’ assessment of parents’ intentions to vaccinate with the adjuvanted seasonal flu vaccine and parents’ actual intentions to vaccinate.

	Clinician Assessment of Parental Intention to Vaccinate with Influenza Vaccine Based on Availability and Cost (N = 207)	
Actual Parental Intention to	Correlation Coefficient	*p* Value
Vaccinate with aTIV “… when it becomes available”	0.600	<0.001
Vaccinate with aTIV if free	0.483	<0.001
Vaccinate with aTIV if $25	0.266	<0.001
Vaccinate with aTIV if $50	0.146	0.036

## Data Availability

The data presented in this study are available on request from the corresponding author.

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
