# Peer review of "Clinicians Are Not Able to Infer Parental Intentions to Vaccinate Infants with a Seasonal Influenza Vaccine, and Perhaps They Should Not Try: Findings from the Pediatric Influenza Vaccination Optimization Trial (PIVOT)–IV"

_vaccines, 2022, doi:10.3390/vaccines10111955_

Round 1

Reviewer 1 Report

This is an interesting study of clinician assessment of parents intention to vaccinate their young children with recommended seasonal flu vaccine.  The survey was conducted in public health and community practice clinics in Canada in 2015 and 2016, six years ago.    Although there is a statistically significant association between clinician perception and parental intention, you note that this accounts for a minority of the variance in parental intentions.

It is not surprising that parents are most inclined to accept the vaccine when there is no cost for them.

These findings can be used to support clinician training in sensitivity and persuasion skills.  You note correctly that others have noted that strength of clinician recommendation is correlated with parental intention.

Results:

Table 1 seems to be missing some information, specifically female n(%).  Also it is unclear what the numbers and %s refer to on the rows "Parents" and "Children". 

Overall, it seems to this reviewer that clinicians do often understand parental intention. It would be helpful to know how well parental intention correlates to receipt of vaccine by the children. 

Author Response

Thank you for your assessment.

Table 1 seems to be missing some information, specifically female n(%). Also it is unclear what the numbers and %s refer to on the rows "Parents" and "Children".

Thank you. The table has been reformatted and headers adjusted for clarity.

Overall, it seems to this reviewer that clinicians do often understand parental intention. It would be helpful to know how well parental intention correlates to receipt of vaccine by the children.

You identify an important topic and investigated/reported in the companion manuscript, PIVOT-I.

Reviewer 2 Report

Comments:
While I agree that correlation is the appropriate method to assess concordance of clinicians’ perceptions and parents’ actual intentions, I
find the Methods section inadequate (and as a consequence presentation
of data and analysis may change):

(1) description of the "scores"
For both variables a 7 point scale has been used; therefore these scores
are ordinal variables.
You may wish to show diagrams to visualize these relationships.

(2) The method of correlation used is not mentioned (Pearson correlation
method or Spearman rank correlation method)
While the Pearson correlation coefficient is the most widely used, it measures the strength of the linear relationship between normally
distributed variables.
When the variables are not normally distributed or the relationship
between the variables is not linear, it may be more appropriate to use
the Spearman rank correlation method (non-parametric test).
When the nominal variable has more than 2 categories, correlation tests
violate the assumption of linearity.
The measures of effect would also be different:
     if correlation linear: Pearsons product-moment correlation coefficiënt.
     if correlation non-linear: Spearmans rho or Kendalls tau

(3) Throughout the paper only correlation coefficients are mentioned,
while for the reader the percentage of the effect (named determination
coefficient, which equals the coefficient squared (R2) ) which is related to this correlation is most related to clinical/public health practice.

(4) The paper uses p-values to indicate significance: "Correlation coefficients and corresponding 2-tailed measures of significance were calculated" [e.g. Correlation coefficient 0.483 (P<0.001)] 
However, in line with (3) above, 95% confidence intervals for determination coefficients (coeff. squared (R2) ) would be more informative to assess the effect.

(5) While for clinicians only 1 clinician per community practice or the public health clinic has been enrolled, the 207 parents are clustered over 15 community practices and 9 public health clinics (with an average of about 8 per cluster).
There may well be little variance within clusters (in the most extreme case of no within-cluster variance, these 207 parents yield as much information as 15+9=24 parents).
You may have to consult an expert statistician to handle this statistically.

(6) There may be a strong difference in parents' willingness to pay between community practices and public health clinics. As a result, also the concordance of clinicians’ perceptions and parents’ actual intentions may differ between these two groups.
While a small sample size in some strata may be a hindrance, no attempt
has been made to analyze this further.
In any case, this should be addressed under "Discussion".

minor comment
     Table 2.
     heading of column 2 should read: N (range) and mention N=207 in the
table title.

Author Response

(1) In our opinion, graphic presentations of the modest correlations observed
would not be informative. Because the relationships of the variables are so
modest graphic presentation of the modest relationships would convey
very little information.
(2) We utilized Pearson correlation analyses which we found were appropriate to the nature and distribution of the variables under
study.
(3,4) In most cases, we do present both the correlation and the proportion of variance accounted (R2) for in our results, and make the point that clinician judgment accounts for little variance in parental vaccination intentions.
(5) While intraclass correlations due to clinician unique characteristics would
potentially be an issue, examination of the relationships observed (data not reported) showed that the pattern and strength of relationships of clinician judgment and parental intention were similar across clinicians and thus an aggregated correlational approach is justified.

Minor Comment

Table 2. heading of column 2 should read: N (range) and mention N=207 in the table title.
This has been revised for clarity and corrected in the manuscript.

Round 2

Reviewer 2 Report

comments of the reviewer have been addressed.